# The Linkage between Ethical Leadership, Well-Being, Work Engagement, and Innovative Work Behavior: The Empirical Evidence from the Higher Education Sector of China

**DOI:** 10.3390/ijerph19095414

**Published:** 2022-04-29

**Authors:** Kan Jia, Tianlun Zhu, Weiwei Zhang, Samma Faiz Rasool, Ali Asghar, Tachia Chin

**Affiliations:** 1School of Management, Zhejiang University of Technology, Hangzhou 310023, China; jiakan@zjut.edu.cn (K.J.); zhutianlun8888@163.com (T.Z.); tachiachin@zjut.edu.cn (T.C.); 2School of Cultural Creativity and Management, Communication University of Zhejiang, Hangzhou 310019, China; 3Dr. Hassan Murad School of Management, University of Management and Technology, Lahore 54770, Pakistan; ali.asghar@umt.edu.pk

**Keywords:** ethical leadership, well-being, innovative work behavior, work engagement, higher education

## Abstract

In this study, we investigate the relationship between ethical leadership (EL), work engagement (WE), well-being, and innovative work behavior (IWB). The significance of these variables has increased in the current era when the influence of technology is exponentially increasing in the education sector. We investigate the role of ethical leadership in determining innovative work behavior. Moreover, we investigate the moderating effect of WB in the relationship between EL and WE. We also examine the mediating impact of WE in the relationship between EL and IWB. We used a questionnaire survey approach to collect data. The target population of this study was the academic personnel, i.e., senior professors, lecturers, and supporting staff associated with the higher education sector located in Zhejiang Province, China. Data were collected in two phases. In the first phase, we sent 300 research questionnaires and received 251 responses. In the second phase, after a three-month interval, we sent 200 questionnaires and received 162 responses. However, over the two phases, we collected a total of 413 questionnaires; 43 were discarded. Therefore, for analysis, we used 370 questionnaires. The data were analyzed using the structural equation modeling through SmartPLS 3.2.2. First, in the direct relationship, results confirm that EL positively influences the IWB. Secondly, WB has a positive and moderating relationship between EL and IWB. Thirdly, we address the relationship between EL and WE. The outcome indicates that there is a positive and significant relationship. Fourth, the results of this study indicate that there is positive and significant relationship between WE and IWB. Finally, the outcomes imply that WE positively mediates between EL and IWB. Ethical leadership and well-being are important for innovative work behavior that supports managers in introducing a supportive workplace environment that promotes good interpersonal relationships with subordinates. Therefore, a good interpersonal relationship between managers and subordinates enhances the work quality. So, ethical leaders provide a supportive work environment to all subordinates regarding their work.

## 1. Introduction

The contemporary political and economic realities require higher education to contribute to national competitiveness by focusing on the employability of graduates above its traditional role of creating informed citizens and improving the well-being of individuals and society at large [1]. According to the United Nation’s sustainable development goals (SDGs), the higher education sector plays a key role in national development [2,3]. In China, the higher education policy has undergone major shifts to reach its current form, where it is highly internationalized [4]. The government of China has competently updated the systems of the higher education sector through the strategic adoption of neoliberal methods for education market formation in the country. In this regard, employee well-being in higher education institutions plays a significant role in enhancing educational institutions’ performance [5]. Such institutions are the hallmark of the Chinese education regime as the ethics of faculty running them include key traditional elements at their core, including loyalty to education, love of life, and selfless dedication [6].

Ethical leadership, specifically in educational institutions, is considered to be driven by values, including the firm belief in the dignity and rights of others instead of personalities or politics [7]. Ethical leadership enhances the education institution’s performance [8]. Likewise, a study conducted by Schwepker Jr and Dimitriou [9] reports that ethical leadership reduces job stress and improves performance quality in the hospitality industry. Ethical leadership also plays a significant role in keeping teams together in a workplace environment as it moderates the effects on team management and work efficiency [10]. Likewise, ethical leadership also plays a moderating role between psychological ownership of knowledge and knowledge hiding [11]. In their recent study, Marquardt, Casper [12] report that the employees perceive their leaders as less ethical when they emphasize avoiding failure while downplaying the importance of personal learning and development. Therefore, managers must foster an ethical workplace by promoting interpersonal and informational justice [13]. 

The positive side of leaders increases workplace engagement among the employees associated with the public and services sector organizations [14]. Workplace engagement, as defined by Schaufeli, Salanova [15], is “a positive, fulfilling, work-related state of mind that is characterized by vigor, dedication, and absorption”. The individuals engaged in their work transform the energy into creativity and organizational commitment [16]. Workplace engagement level is high among the employees in their initial days of joining an organization, whereas it stabilizes over time [17]. Workplace engagement has multiple predictors. For example, higher resilience, job satisfaction, and lower morale distress result in increased workplace engagement [18]. The contributors to work engagement include physical, mental, psychological, and cultural engagement with the organization, whereas its consequences include job satisfaction, innovation, and leadership development [19]. 

Employee well-being refers to mental and emotional health, positive attitude, and job satisfaction in an organizational setting [20]. Although it is not a new concern, it is considered a broader issue in contemporary organizations. Employee well-being generally refers to improving their health in terms of work-related safety [21]. The concept of employees’ well-being is not limited to health and safety, it also incorporates physical activity, job satisfaction, and personal development [22]. According to ILO [23], “the purpose of measuring employee well-being is to complement occupational and safety health measures to make sure workers are safe, healthy, satisfied and engaged at work”. Employee well-being is predicted by authentic leadership, which fosters workplace engagement and innovative work behavior in the employees [24]. 

Employees exhibit complex behavior to generating, introducing, and applying innovative ideas at their workplace. Innovative workplace behavior gives competitive advantage and sustainability to the respective organization [25]. Previous studies show that innovative workplace behavior is negatively associated with workplace violence [26], workplace ostracism, and workplace incivility [27]. Work engagement and supervisor support significantly predict innovative workplace behavior [28]. Innovative workplace behavior significantly maintains university services’ sustainability and increases industries’ competitiveness [29]. The literature shows that innovative workplace behavior among university faculty members can be cultivated through authentic leadership practices and psychological capital management [30].

The aim of this study is to investigate the relationship between ethical leadership, WE, well-being, and IWB. The significance of these variables has increased in the current era when the influence of technology is exponentially increasing in every field of life. In addition, the COVID-19 pandemic has cast lasting effects on everyday life, especially on work-life [31,32]. The time calls for an exploration of the effects of ethical leadership on work engagement and, eventually, on innovative work behavior [33]. Despite the increased emphasis on ethical leadership, WE, well-being, and IWB, studies focusing on these variables were rarely found. At the same time, the extant literature discusses the importance of ethical leadership, well-being, work engagement, and innovative work behavior [34,35,36]. Most of the researchers have focused on cases where employee well-being and innovative work behavior is exercised in developed countries such as Europe, USA, and Singapore. However, research into the collateral relationship between ethical leadership, well-being, work engagement, and innovative work behavior is still in its infancy and has not provided meaningful results. Only a few studies have examined the direct relationship between ethical leadership and work engagement or between well-being and innovative work behavior [37,38]. However, the relationship between ethical leadership, well-being, work engagement, and innovative work behavior is still unexplored. Especially considering well-being as an intervening construct and work engagement as a moderating variable still needs to be researched. Based on the above-mentioned potential research impetus, we develop a conceptual framework (Figure 1) of this study and address the following research questions (RQ):RQ1.Does ethical leadership influence work engagement and innovative work behavior?RQ2.Does well-being moderate between ethical leadership and work engagement?RQ3.Does work engagement intervene between ethical leadership and innovative work behavior?

## 2. Theory and Hypotheses Development

### 2.1. Ethical Leadership and Innovative Work Behavior

Several studies explored the phenomena of EL in the context of IWB [39,40,41,42]. Ahmad, Gao [39] point out that EL effectively enhances IWB in workers with less proactive personalities. They also found that psychological safety and interaction between the leaders and followers mediate this relationship. Ullah, Mirza [40] observed that ethical leadership positively influences innovative work behavior, while human capital and social capital mediate their relationship. Haque and Yamoah [42] conducted a comparative study between Canada and Pakistan and suggested that supportive leaders encourage subordinates that affect their creative behavior in the workplace. They found that ethical leadership reduces occupational stress and increases innovative work behavior. They report that the effect of EL on IWB is higher among Pakistani workers as compared to their Canadian counterparts. Ullah, Mirza [43] found that ethical leadership fosters innovative work behavior in employees, while social capital mediates this relationship. Zaman, Wang [44] found that leadership indirectly affects innovative work behavior through psychological capital. So, we proposed the first hypothesis as follows: 

**H1:** *Ethical leadership positively impacts innovative work behavior*.

### 2.2. Ethical Leadership and Work Engagement

The relationship between ethical leadership and work engagement seems to be getting the attention of researchers in the current era [45,46,47]. Alam, Fozia [48] found that ethical leadership has a positive impact on organizational commitment, with the mediating role of work engagement. Fuller [45] noted that there is a positive relationship between ethical leadership and work engagement. Work engagement has a strong relationship with compassion satisfaction as the teachers exhibit their engagement through care towards their students. Although ethical leadership promotes work engagement, it is not mediated by compassion satisfaction [47]. Ethical leadership exerts positive effects on work engagement and workaholism. However, these relations are not moderated by self-efficacy [46]. The relationship between EL and WE, well-being and firm performance is mediated by ethical culture in Pakistan and Italy. In response to ethical leadership, Pakistani employees showed higher work engagement, while Italian employees exhibited higher levels of well-being [49]. The quality of leadership plays a significant role in the significant connection between EL, well-being, and WE [50]. Based on the above-discussed literature on ethical leadership and work engagement, the present study proposes the following hypothesis for empirical testing:

**H2:** *Ethical leadership positively impacts work engagement*. 

### 2.3. Moderating Effect of Well-Being

Several authors have studied ethical leadership as an antecedent of the well-being of employees [51]. Ethical leadership is positively associated with trust in management and the psychological well-being of the employees [52]. The relationship between ethical leadership and employee well-being is completely mediated by organizational citizenship anxiety [53]. Employee well-being strongly occurs as a reaction to unethical behavior rather than a consequence of ethical behavior [54]. Teimouri, Hosseini [55] have also found a positive and significant relationship between ethical leadership and employee well-being [56]. Therefore, providing significant evidence between the connection of EL and well-being, the literature also indicates the relationship between well-being and workplace engagement. Work engagement partially mediates the relationship between psychological well-being and employees’ job performance [57]. Zeng, Chen [58] found a positive relationship between well-being and work engagement among secondary school teachers in central China. Xu, Xie [59] also reported a positive relationship between affective well-being and work engagement of the employees. Likewise, Sarwar, Ishaq [49] also argue that EL has a significant relationship with well-being and work engagement. Considering the review of the contemporary relevant literature, the present study develops the following hypothesis for further testing and validation:

**H3:** *Well-being positively moderates between ethical leadership and work engagement*.

### 2.4. Work Engagement and Innovative Work Behavior

The meta-analysis of contemporary literature shows that WE has a medium to large correlation effect on IWB [60]. The innovative work behavior of employees is affected by their regulatory focus. Work engagement improves employees’ innovative work behavior by having either a promotion or prevention regulatory focus [61]. Besides being a predicator of innovative work behavior, the work engagement of employees also indicates a learning organization [62]. The WE of the workers brings creativity and novelty in the routine work [63]. Afsar, Al-Ghazali [64] found that WE and interpersonal trust mediate the effect of cultural intelligence on IWB. Further, Montani, Vandenberghe [65] discovered that WE plays a mediating role between workload and innovative work behavior, provided that the workload is moderate. They also found that WE mediated between workload and IWB. It was found that WE and IWB are positively related, provided necessary resources are available for them [66]. Further, the studies of Van Zyl, Van Oort [67] and [68,68] also confer the relationship between WE and IWB. Based on the review of literature, the present study poses the following hypothesis for testing:

**H4:** *Work engagement positively impacts innovative work behavior*.

### 2.5. Mediating Effects of Work Engagement 

The contemporary literature shows an increasing number of studies addressing the mediating role of WE on IWB with reference to several other variables [38]. The study conducted by Asif, Qing [36] reports a positive relationship between ethical leadership, WE, and IWB. Work engagement mediates the relationship between leadership and job performance [69]. Li, Sajjad [68] argue that work engagement plays a mediating role in the relationship between leadership and innovative work behavior. Figure 1 presents the comprehensive conceptual framework of this study. Thus, we propose the fifth and last hypothesis as follows:

**H5:** *Work engagement positively mediates between ethical leadership and innovative work behavior*.

## 3. Research Methods

### 3.1. Research Approach

We used a quantitative research approach to collect and analyze the data [13]. The data were collected through a questionnaire survey. There are two main reasons to apply the questionnaire survey technique [27]. First, researchers can collect maximum data in a minimum time using this approach. Second, this approach is not expensive as compared to other research approaches [44]. 

### 3.2. Instrument Designing 

The instrument (questionnaire) was drafted in the English language, but later on, translation was conducted into the Chinese language. Then, the authors conducted an experimental study (pilot study) of the instrument. The respondents of this experimental study were ten doctorate students and ten academic professors. Moreover, the respondents were familiar with the research topic and both languages. These respondents suggested some changes to the instrument. Therefore, the instrument was revised as per the feedback of the respondents. After finalizing the research instrument, they were disseminated among the target population.

### 3.3. Data Collection and Sampling

The target population of this study was academic personnel, i.e., senior professors, lecturers, and supporting staff associated with the higher education sector located in Zhejiang Province, China. The ethics committee reviewed the research. As regards research ethics, the authors informed respondents that their data would remain confidential and be used only for research purposes. The survey was conducted through WJX (https://www.wjx.cn), which is a Chinese-based website that is well-known in China for data collection. WJX is a third-party website that helps to gather data from the given sample. A multilevel approach was adopted to collect the data to avoid common method bias. The questionnaire comprised 19 items related to ethical leadership, well-being, employee, innovative work behavior, and work engagement. Data were collected in two phases. In the first phase, we were sent 300 research questionnaires and received 251 responses. In the second phase, after a three-month interval, we sent 200 questionnaires and received 162 responses. However, over the two phases, we collected a total of 413 questionnaires; 43 were discarded due to missing values. Therefore, for analysis, we used 370 questionnaires.

### 3.4. Variables and Measures

We used ethical leadership as an independent variable and innovative work behavior as a dependent variable. Similarly, we used well-being as a moderating variable and employee engagement as a mediating variable. The items of these variables were adopted from the existing studies [26,27,36,70]. 

#### 3.4.1. Ethical Leadership 

The three items of ethical leadership were taken from Asif, Qing [36]. The sample items of ethical leadership were “My supervisor disciplines employees who violate ethical standards” and “My supervisor sets an example of how to do things the right way in terms of ethics”. According to Joseph F Hair, Ringle, and Sarstedt (2013), the acceptable standard value of Cronbach alpha value must be greater than 0.7. However, the alpha value of ethical leadership was 0.648, which is an acceptable standard. 

#### 3.4.2. Innovative Work Behavior 

The six items of innovative work behavior were adopted and modified from existing studies [26,27]. The innovative work behavior is represented by items such as: “I feel that my I am more efficient than my supervisor/co-worker/subordinate”, “During the past six months, my actual work creativity is increasing day by day”. However, the alpha value of the innovative work behavior was 0.803, which is an acceptable standard.

#### 3.4.3. Well-Being 

The four items of well-being were adopted and modified from existing studies [70]. The well-being construct was represented by items such as “Do you feel that your boss is empathic and understanding about your work concerns”, “Do people at your work believe in the worth of the organization”. However, the outcome indicates that the alpha value of ethical leadership is 0.714, which is an acceptable standard. 

#### 3.4.4. Work Engagement 

We used seven items to measure work engagement. The scales used in this research were adapted from [70]. The sample items of work engagement were “I am committed to continuous quality improvement in my work”, “My supervisor positively motivates my performance at work”. The results confirmed that the alpha value of work engagement is 0.848, which indicates that all the items were reliable and valid. 

### 3.5. Demographics 

The respondents’ demographics are given in Table 1. In terms of the gender of the participants, 42% of them were female, and 58% of them were male. The data gathered from private universities made up 44% of the total, and 56% were from public sector universities; 36% of respondents were senior professors, 38% were lecturers, and 26% were supporting staff. Similarly, in terms of the respondents’ education, 39% were doctorate degree holders, and 61% of respondents held postgraduate degrees.

## 4. Data Analysis

### 4.1. Reliability and Validity

The data were analyzed using the structural equation modeling through SmartPLS 3.2.2. This method is useful in this study for three reasons; first, researchers [1,2] suggested that PLS-SEM is the best technique for exploring the theory and the relation of new variables. The findings of our research found the mediating effect of WE between EL and IWB. Similarly, we also found a moderating effect of well-being between EL and WE, which leads us to find also a mediating and moderating effect of WE between well-being and IWB. Second, CB-based PLS-SEM does not need a large amount of data to run structural question modeling [3]. Therefore, this research collected a moderate number of samples. Third, researchers suggest that PLS-SEM is useful for complex multivariate analysis [4]. This study has assessed the cause-effect, mediation analysis, and mediated moderation analysis for the set conceptual framework. A two-step analysis procedure helped in data analysis [5]. 

First of all, we assessed the reliability and validity of the constructs. The consistency of the constructs measured the item loading with their relevant factors. The item loadings were arranged from 0.605 to 0.848. Researchers [6] suggested that the minimum item loading should be 0.7. Hence, all indicators have shown their item loadings above the threshold of 0.7. Therefore, items were consistent with their relevant constructs. Cronbach alpha, rho alpha, composite reliability, and average variance extracted measured the consistency and reliability of the constructs. Researchers [7,8] suggested that the value of Cronbach alpha, rho alpha, and composite reliability should be above the threshold of 0.7. The Cronbach alpha values for all constructs were above the threshold of 0.7. The rho alpha and composite reliability values for all constructs were also above 0.7. Therefore, the variable under study was consistent and reliable. At the same time, the average variance extracted values above the threshold of 0.5, according to the researchers’ recommendation [9]. Hence, all constructs were reliable and valid, as given in Table 2. 

Researchers [10] have introduced a new procedure of Heterotrait-Monotrait ratio of correlations (HTMT) to measure discriminant validity. The HTMT values are used to measure discriminant validity, which is one of the essential parameters to assess model quality. Researchers [11] have suggested that HTMT values should be below 0.9. We observed that HTMT values for all variables were less than 0.9. Hence, all first-order factor analysis has shown a satisfactory level of HTMT ratios, as given in Table 3. 

### 4.2. Multicollinearity

Variance inflation factor (VIF) is another measure to assess multicollinearity issues among constructs. VIF values must be below the threshold of 5 [8,12]. All constructs have shown an inner VIF value below the threshold of 3, which shows that there was no multicollinearity issue. PLS-SEM uses Normed Fit Index (NFI) and Standardized Root Mean Square Residual (SRMR) indices to measure the fitness of the SEM model. Researchers [13] suggested that SRMR value must be below the threshold of 0.06, and the NFI value must be above the threshold of 0.8 [14]. The model has shown NFI = 0.911 and SRMR = 0.044, which shows an adequate fitness of the model, as given in Table 4. 

### 4.3. Goodness of Fit 

The goodness of fit (GoF) shows the effectiveness of the model based on quantitative data [15]. The range of GoF is from 0 to 1 (0.36 = effective, 0.25 = average, below 0.1 is weak). The goodness of fit index shows the plausibility and parsimony of the model. The formula of GoF is “GoF = sqrt ((average AVE) ∗ (average R^2^))”. The GoF value of 0.51 shows that the model was parsimonious and plausible, as given in Table 5. 

The f-square value shows the effect of exogenous construct on endogenous constructs [16]. An effect size below 0.02 is considered weak, above 0.15 moderate and above 0.35 as substantial. The moderator (EL*WB) has shown a minimum level of effect size (f^2^ = 0.024) on WE, while well-being has also shown a substantial effect (f^2^ = 0.317) on WE. EL has also shown a moderate effect on IWB (f^2^ = 0.216) and a substantial effect on WE (f^2^ = 0.524). The WE has also shown a minimum effect level on IWB (f^2^ = 0.027), as shown in Table 6. 

Coefficient of determination R-square indicates change in the dependent construct due to the per unit change in the independent construct. It must be above a threshold of 0.1 [5,17]. Innovative Work Behavior has shown 33% and Work Engagement has shown 52% power of prediction in a SEM model, as given in Table 7.

### 4.4. Hypotheses Testing 

#### 4.4.1. Direct and Indirect Effects

One researcher [18] suggests that mean values in PLS paths are the same as beta values in regression analysis. Beta measures the per unit change independent variable due to the independent variable. The value of beta was windows with the help of a significant value or probability. Another measure to endorse the beta value was T-test. We tested the hypothesis with the help of beta values and T-test as well as bootstrapping at subsample level 5000. Data showed that gender and program have no effect on the dependent variable of cognitive presence with a *p*-value greater than 0.05. The findings demonstrate that EL has a positive connection with IWB (M = 0.468, *p* < 0.001). So, the first hypothesis is accepted. Similarly, EL has a positive connection with WE (M = 0.530, *p* < 0.001). Hence, it leads us to not reject the hypothesis. The results also show that well-being has a positive and significant effect on WE (M = 0.411, *p* < 0.001). It leads us to not reject the hypothesis. Work engagement has also shown a positive and significant effect on IWB (M = 0.165, *p* < 0.001). It leads us to not reject the hypothesis. Work engagement has shown a mediation between EL and IWB (M = 0.087, *p* < 0.001). It leads us to accept the robust hypothesis as given in Table 8. Figure 2 also shows all possible direct relations among constructs. 

#### 4.4.2. Moderation and Mediated Moderation

We tested the moderation and meditated moderation, and the findings demonstrate that well-being has a positive connection with WE (M = 0.411, *p* < 0.001). It leads us to measure the moderating effect of EL*WB on WE, which was found to be positive and significant (M = 0.078, *p* < 0.001). Hence, the hypothesis was accepted. The nature of the moderation is shown in Figure 3. The mediated moderated effect of WE between EL*WB and IWB was also found to be significant and positive (M = 0.013, *p* < 0.001). So. the hypothesis was not rejected. However, Figure 3 shows the moderating effect of well-being with the EL and WE. The moderation and mediated moderation detail are also given in Table 9.

## 5. Discussion

In this study, we examine the role of ethical leadership in determining innovative work behavior. Moreover, we investigate the moderating effect of well-being in the relationship of ethical leadership and employee engagement. We also investigate the mediating role of WE in the connection between EL and IWB.

First, we found that EL is positively associated with IWB, which supports the first hypothesis. Samma, Zhao [27] conducted a study of workplace violence and innovative work behavior. The results of their study indicate that supportive leaders motivate employees, which enhances the morale of employees and brings creativity to their work behavior.

Secondly, we explore the connection of ethical leadership with work engagement, and the findings of our research demonstrate that ethical leadership boosts work engagement. Zhou, Rasool [26] conducted a study among the Chinese workers of SMEs, and the results of their study also show that supportive leaders motivate their followers. So, the followers follow the instruction of the leaders that connects them with their work, goals, mission, and organizational commitment.

Third, we test the moderating effect of well-being in the connection between ethical leadership and work engagement. The findings confirmed that well-being positively moderates between ethical leadership and work engagement. The finding of our study is also supported by Iqbal, Qureshi [71]. They conducted a study among a Pakistani organization. Their study shows that ethical leaders work for the employees’ well-being and emotionally engage them with their tasks, converting them into creative workers. Therefore, it has been proven that this research supports our outcomes.

Fourth, we suggest that the employees’ work engagement positively affects their creative work behavior. However, our results confirmed that WE has a positive impact on IWE. Some previous studies also support our findings. Rasool conducted a study on the relationship between the workplace environment and sustainable work performance. Their study points out that work engagement develops a bridge from regular work practices to creative work.

Finally, in this study, we explore the mediating effect of work engagement on the relationship between ethical leadership and innovative work behavior. Wang, Rasool [72] conducted a study to test the relationship between despotic leadership and the success of the projects, and their results suggest that through work engagement, ethical leaders motivate the workers to feel passionate about their tasks, and loyal to the organization, and put flexible energy to bring the creativity and innovativeness in their work.

## 6. Conclusions

This study has shown that ethical leadership positively impacts the innovative work behavior of the employees associated with higher education in China. Similarly, the outcomes indicate that the well-being of the employees working in China’s higher education positively moderates between ethical leadership and work engagement. Results also note that ethical leadership has a positive impact on work engagement. Similarly, this study concludes that work engagement brings innovative work behavior that affects the individual and overall organizational performance. Finally, this study confirms that work engagement positively mediates between ethical leadership style and IWB. So, this study confirms that a leader who has ethical values and principles works for the well-being of the employees that engage the employees with work. Ultimately, it affects the individual’s creativity. Ethical leadership is important for innovative work behavior that supports managers in introducing a supportive workplace environment that promotes good interpersonal relationships with subordinates. Therefore, a good interpersonal relationship between managers and subordinates enhances the work quality. So, ethical leaders provide a supportive work environment to all subordinates regarding their work.

These findings have practical implications for leaders and managers associated with the education sector. Moreover, it will provide relevant and meaningful guidance for the managers and educational institutions. First, the people working in the education sector have to have innovative behavior because they must keep up to date with new knowledge in their relevant discipline. The educational leaders should build a system under which their followers can control their jobs and update their knowledge. Secondly, the teachers and support staff should give feedback to their leaders if they are facing any issues on the job. In this way, ethical leaders will work for the well-being of the employees. Third, the educational institution needs to formally organize the professional development training for the leaders, managers, and support staff; this training will bring creativity to their behavior.

## Figures and Tables

**Figure 1 ijerph-19-05414-f001:**
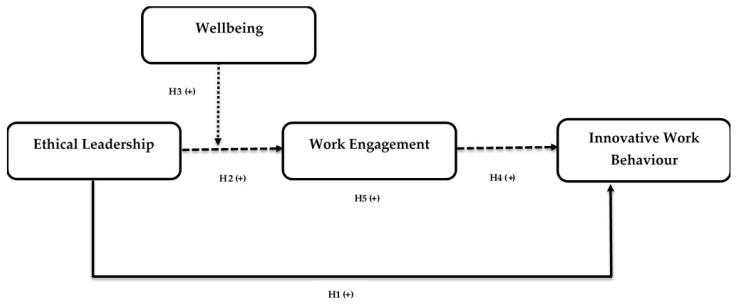
Conceptual Framework. Note: Solid lines show the direct relationships and dotted lines show the moderating and mediation relationships.

**Figure 2 ijerph-19-05414-f002:**
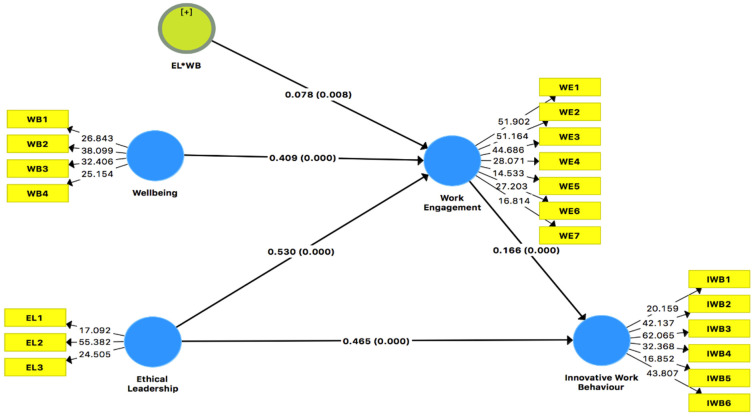
Path Analysis.

**Figure 3 ijerph-19-05414-f003:**
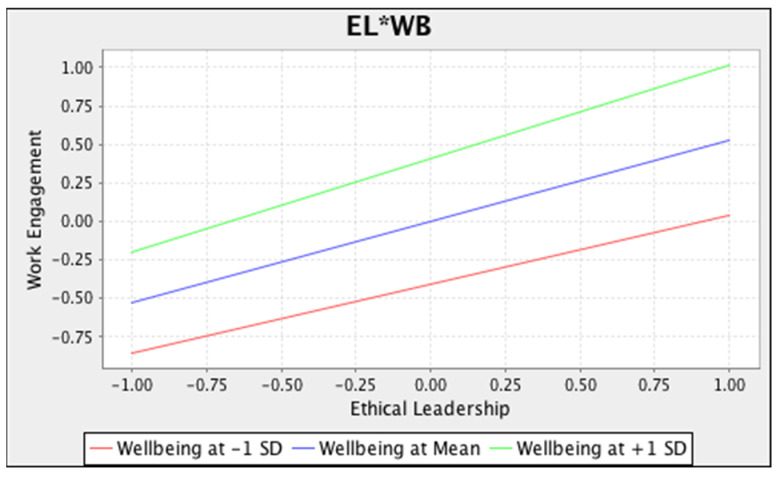
Moderation of EL*WB on Work Engagement.

**Table 1 ijerph-19-05414-t001:** Sample Distribution.

Characteristics		F	%
Gender	Female	156	42
	Male	214	58
Universities	Private	163	44
	Public	207	56
Positions	Senior Professors	133	36
	Lecturers	141	38
	Supporting Staff	96	26
Education	Doctorate	145	39
	Post-Graduate	225	61

**Table 2 ijerph-19-05414-t002:** Reliability and validity.

	Item	Loadings	∝	rho_A	CR	AVE
**Ethical Leadership**	EL1	0.648	0.599	0.66	0.78	0.546
	EL2	0.848				
	EL3	0.706				
**Innovative Work Behavior**	IWB1	0.605	0.803	0.826	0.857	0.503
	IWB2	0.738				
	IWB3	0.817				
	IWB4	0.713				
	IWB5	0.602				
	IWB6	0.756				
**Well-Being**	WB1	0.716	0.714	0.721	0.821	0.534
	WB2	0.764				
	WB3	0.744				
	WB4	0.698				
**Work Engagement**	WE1	0.773	0.846	0.867	0.883	0.52
	WE2	0.825				
	WE3	0.764				
	WE4	0.733				
	WE5	0.611				
	WE6	0.697				
	WE7	0.62				

**Table 3 ijerph-19-05414-t003:** HTMT Ratios.

	Ethical Leadership	Innovative Work Behavior	Well-Being	Work Engagement
Ethical Leadership				
Innovative Work Behavior	0.72			
Well-Being	0.425	0.257		
Work Engagement	0.749	0.472	0.654	

**Table 4 ijerph-19-05414-t004:** Good fit model and VIF.

	Innovative Work Behavior	Work Engagement	Model Fit Indices
Work Engagement	1.497		SRMR = 0.024 NFI = 0.90
Ethical Leadership	1.497	1.116	
Well-Being		1.101	

**Table 5 ijerph-19-05414-t005:** The goodness of Fit Index.

Constructs	AVE	R-Square
Ethical Leadership	0.546	
Well-Being	0.534	
Innovative Work Behavior	0.53	0.330
Work Engagement	0.52	0.518
	0.52	0.424
GoF	0.51	

**Table 6 ijerph-19-05414-t006:** Effect size f-square.

	Innovative Work Behavior	Work Engagement
EL*WB		0.024
Well-Being		0.317
Ethical Leadership	0.216	0.524
Work Engagement	0.027	

**Table 7 ijerph-19-05414-t007:** R-square.

	R Square	R Square Adjusted
Innovative Work Behavior	0.332	0.33
Work Engagement	0.52	0.518

**Table 8 ijerph-19-05414-t008:** Direct and Indirect Paths.

Hypothesis	M	SD	T Stats	*p* Values	Status
Ethical Leadership → Innovative Work Behavior	0.468	0.043	10.857	0.000	Not Rejected
Ethical Leadership →Work Engagement	0.530	0.025	20.841	0.000	Not Rejected
Well-Being → Work Engagement	0.411	0.030	13.675	0.000	Not Rejected
Work Engagement→ Innovative Work Behavior	0.165	0.044	3.769	0.000	Not Rejected
Ethical Leadership → Work Engagement → Innovative Work Behavior	0.087	0.022	3.900	0.000	Not Rejected

**Table 9 ijerph-19-05414-t009:** Moderation and mediated moderation.

Hypothesis	M	SD	T Stats	*p* Values	Status
Well-Being → Work Engagement	0.411	0.030	13.675	0.000	Not Rejected
EL*WB → Work Engagement	0.078	0.030	2.602	0.008	
EL*WB → Work Engagement → Innovative Work Behavior	0.013	0.005	2.415	0.016	Not Rejected

Note: EL = Ethical Leadership; WB = Well-Being.

## Data Availability

The data that support the findings of this study are available from the corresponding author upon reasonable request.

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
