# Peer review of "The Linkage between Ethical Leadership, Well-Being, Work Engagement, and Innovative Work Behavior: The Empirical Evidence from the Higher Education Sector of China"

_ijerph, 2022, doi:10.3390/ijerph19095414_

Round 1

Reviewer 1 Report

This study investigates the Linkage between Ethical Leadership, Well-being, Work Engagement, and Innovative Work Behavior. The topic and theoretical conceptualization is novel. Besides, this study used modern techniques for analysis, which is widely accepted. However, I accept and suggest some minor revisions. The required revisions are given below.

  • Title: It is suggested that the revised the title of the paper. The revised title should be “The Linkage between Ethical Leadership, Well-being, Work Engagement, and Innovative Work Behavior: The Empirical Evidences from the Higher Education Sector of China”.
  • Abstract: It is suggested that the author explain more about the research methods in the abstract.
  • Introduction: The introduction of this study needs to be revised. I suggest the author explain in more details about the aims of the study.
  • Research Methods: It is suggested to the author to describe in more detail about the research approach did the authors used in this study?
    Results: This study used modern techniques for analysis, which is widely accepted in the current research era specifically in the survey analysis approach. The analysis authors performed in this study are well explained.

Discussion: The discussion section is also well explained.

Conclusion: The conclusion section is written well, but it needs more explanation for academic and practitioners’ point of view.

Author Response

Comment: 1. This study investigates the Linkage between Ethical Leadership, Well-being, Work Engagement, and Innovative Work Behavior. The topic and theoretical conceptualization is novel. Besides, this study used modern techniques for analysis, which is widely accepted. However, I accept and suggest some minor revisions. The required revisions are given below.

Response: Thank you very much for your positive and constructive comments and suggestions on our manuscript. We have studied your comments carefully and have revised them. The reviseded part of the manuscript is highlighted in red color.

Comment: 2. Title: It is suggested that the revised the title of the paper. The revised title should be “The Linkage between Ethical Leadership, Well-being, Work Engagement, and Innovative Work Behavior: The Empirical Evidences from the Higher Education Sector of China”.

Response: Thank you for your comments. As per the suggestions of the reviewers we have revised the title of the paper.

Comment: 3. Abstract: It is suggested that the author explain more about the research methods in the abstract.

Response: Thank you for your suggestion. As per the suggestions of the reviewers we have further explain the research methods in the abstract. 

Comment: 4. Introduction: The introduction of this study needs to be revised. I suggest the author explain in more details about the aims of the study.

Response: Thank you for your comment. In response to your comment, we have addressed the aims of the study in the introduction. Please see the last paragraph before research questions in the introduction.

Comment: 5. Research Methods: It is suggested to the author to describe in more detail about the research approach did the authors used in this study?

Response: Thank you for your comments. In response to your comments, we have explained in detail about the research approach that we used in this study. Please see the first sub-heading (3.1) with the title of the research approach. In this heading we have explain the research approach.

Comment: 6. Results: This study used modern techniques for analysis, which is widely accepted in the current research era specifically in the survey analysis approach. The analysis authors performed in this study are well explained.

Response: Thank you very much.

Comment: 7. Discussion: The discussion section is also well explained.

Response: Thank you very much.

Comment: 8. Conclusion: The conclusion section is written well, but it needs more explanation for academic and practitioners’ point of view.

Response: As suggested by the reviewer we have edited the conclusion part of the study. We hope that the revised implications are addressed the raised concerns.

We would like to express our great appreciation to you and your comments on our paper.

Thank you and best regards.

Reviewer 2 Report

The article is well written, the research method is properly described, the conclusions are well-founded. I have three following remarks.

Page 2: “Marquardt, Casper [12] report that the employees perceive their leaders less ethical when they emphasize avoiding failure while downplaying the importance of personal learning and development. (raise gap)” What does "raise gap" refer to?

The first paragraph of page 6: The sample items of ethical leadership were: The Cronbach alpha represents the reliability of the construct.  The sample items are “My supervisor (…).” The sentence „The Cronbach …”should be moved further. „According to Joseph F Hair, Ringle, and Sarstedt (2013)”  - the quote should be in square brackets and numbered in the order in which it appears.

Author Response

Comment: 1. The article is well written, the research method is properly described, the conclusions are well-founded. I have three following remarks.

Response: Thank you for your comments on our manuscript. We have addressed all of your comments and have made significant revisions to the manuscript. We feel that our revisions strengthen the manuscript considerably from the original submission. I also want to bring into reviewer’s kind notice the revision part of the manuscript is highlighted in red color.

Comment: 2. Page 2: “Marquardt, Casper [12] report that the employees perceive their leaders less ethical when they emphasize avoiding failure while downplaying the importance of personal learning and development. (raise gap)” What does "raise gap" refer to?

Response: Thank you very much for identifying the typo mistake. It was a typo mistake.  In response to your comment, we have removed the word “raised gap”.

Comment: 3. The first paragraph of page 6: The sample items of ethical leadership were: The Cronbach alpha represents the reliability of the construct.  The sample items are “My supervisor (…).” The sentence „The Cronbach …”should be moved further. „According to Joseph F Hair, Ringle, and Sarstedt (2013)”  - the quote should be in square brackets and numbered in the order in which it appears.

Response: Thank you for your comments. It was also a typo mistake but in the revised version we have rectify the sentences.

.

We would like to express our great appreciation to you and your comments on our paper.

Thank you and best regards

Reviewer 3 Report

Dear Authors,

My comments are within the pdf file.

Author Response

Comment: 1. In this research, the authors investigate the role of ethical leadership (EL) in determining innovative work behavior (IWB). Moreover, they examine the moderating effect of well-being in the relationship between ethical leadership and work engagement. They also investigated the mediating impact of work engagement in the relationship between EL and IWB. They collected data using the research questionnaire from senior professors, lecturers and supporting staff associated with the universities in Zhejiang Province, China. This paper is a well-argued and valuable contribution by analyzing the variability of healthcare expenditures on users with similar scores classified by the Adjusted Morbidity Group (AMG) in Spain.

Response: Thank you for your comments on our manuscript. We have addressed all of your comments and have made significant revisions to the manuscript. We feel that our revisions strengthen the manuscript considerably from the original submission. I also want to bring into reviewer’s kind notice the revision part of the manuscript is highlighted in red color.

Comment: 2. The paper is not structured well, and ignores the theoretical contribution of the research. Moreover, it is written poorly, and the quality of the English language is unsatisfactory. In my opinion, the article needs to be restructured considering the following observations:

Response: Thank you very much for your comments. For the satisfaction of the reviewer we have provide the more rational background of the theoretical contributions in the introduction as well as in the conclusion part.

Moreover, we want to bring into reviewer kind notice that we have taken the English editing services by the professional English editing professional. Please find the attached English editing certificate with the report.

Comment: 3. Abstract: We investigate the role of ethical leadership to in determining…The results conclude. (incomplete sentence)

Response: Thank you for your comment. The sentences are revised as suggestions.

Comment: 3. Introduction.

  • higher education sector plays a key role for in national development
  • the rise of minban (people-run) higher education institutions have has played
  • Ethical leadership significantly effects affects the learning
  • Organizational leadership influence influences o the Covid-19 pandemic has casted cast
  • Response: Thank you very much for your comments. We have addressed the raised concerns.

Reviewer 4 Report

  • Ethical leadership, well-being, work engagement and innovative work behavior are four important scientific categories, which may be explored separately. Relationships among them are very complicated and for sure worth to analyzing. Authors decided to do this, what is one of the prons of he article.
  • The article presents issue adequate to the aims and scope of the IJERPH.
  • The article possesses both, scientific and practical value.
  • The abstract reflects the main topic of the study.
  • The keywords are adequate to the issues presented in the study.
  • The research design is appropriate, methods are clearly exlained.
  • The essential characteristics of the sample have been adequately described.
  • The results are credible.
  • The bibliography doesn't raise any objections.
  • There are reservations about the way in which the key terms contained in the title are used. In one place the authors use full names, in the second abbreviations. This should be harmonized. The best will be to explain at the beginning which term corresponds to a certain abbreviation and stick to this rule unil the end of the article, using one form or the second. 

Author Response

Comment: 1. Ethical leadership, well-being, work engagement and innovative work behavior are four important scientific categories, which may be explored separately. Relationships among them are very complicated and for sure worth to analyzing. Authors decided to do this, what is one of the prons of he article.

Response: Thank you for your comments on our manuscript.

Comment: 1. The article presents issue adequate to the aims and scope of the IJERPH.

Response: Thank you very much.

Comment: 2. The article possesses both, scientific and practical value.

Response: Thank you very much.

Comment: 3. The abstract reflects the main topic of the study.

Response: Thank you very much.

Comment: 4. The keywords are adequate to the issues presented in the study.

Response: Thank you very much.

Comment: 5. The research design is appropriate, methods are clearly explained.

Response: Thank you very much.

Comment: 6. The essential characteristics of the sample have been adequately described.

Response: Thank you very much.

Comment: 7. The results are credible.

Response: Thank you very much.

Comment: 8. The bibliography doesn't raise any objections.

Response: Thank you very much.

Comment: 8. There are reservations about the way in which the key terms contained in the title are used. In one place the authors use full names, in the second abbreviations. This should be harmonized. The best will be to explain at the beginning which term corresponds to a certain abbreviation and stick to this rule unil the end of the article, using one form or the second.

Response: Thank you very much. As per the suggestions of the reviewer we have addressed the raised concern.

Round 2

Reviewer 3 Report

Dear Authors,

I congratulate you for improving the article; now it is in a better position to be published.

Regards